# NCTM: A Novel Coded Transmission Mechanism for Short Video Deliveries

## ABSTRACT

With the rapid popularity of short video applications, a large number of short video transmissions occupy the bandwidth, placing a heavy load on the Internet. Due to the extensive number of short videos and the predominant service for mobile users, traditional approaches (e.g., CDN delivery, edge caching) struggle to achieve the expected performance, leading to a significant number of redundant transmissions. In order to reduce the amount of traffic, we design a Novel Coded Transmission Mechanism (NCTM), which transmits XOR-coded data instead of the original video content. NCTM caches the short videos that users have already watched in user devices, and encodes, broadcasts, and decodes XOR-coded files separately at the server, edge nodes, and clients, with the assistance of cached content. This approach enables NCTM to deliver more short video data given the limited bandwidth. Our extensive trace-driven simulations show how NCTM reduces network load by 3.02%-14.75%, cuts peak traffic by 23.01%, and decreases rebuffering events by 43%-85% in comparison to a CDN-supported scheme and a naive edge caching scheme. Additionally, NCTM also increases the user's buffered video duration by 1.21x-13.53x, ensuring improved playback smoothness.

## CCS CONCEPTS

• **Networks → Mobile networks**.

## KEYWORDS

short video delivery, coded transmission, client-side cache

## 1 INTRODUCTION

Short video applications such as Tiktok [1] and YouTube Shorts [2] are rapidly rising in popularity, attracting billions of active users per month [3, 4], and consistently topping the best-selling apps lists [5]. Taking Tiktok as an example, it had 1.4 billion monthly active users in 2022, and it is predicted to reach 1.8 billion by the end of 2023 [6]. Billions of users imply a huge number of video streams. According to TikTok's report [6], globally, 167 million hours of short videos are consumed every minute, putting an enormous pressure on the current Internet infrastructure.

Traditional video transmission solutions maintain videos on the cloud and stream them to users via content delivery networks (CDNs) [7, 8]. A major weakness of CDN is the huge redundant traffic, causing network pressure and affecting the user viewing experience. Edge caching approaches [9–11] can reduce the amount of redundant traffic by caching user-desired contents at edge nodes [12–14]. However, in short video services, users have different preferences determined by their hobbies, culture, educational backgrounds, lifestyles, etc. Even though videos are rarely watched in groups, they might be popular with other users [15]. Therefore, at the edge, it is difficult to identify which are the most popular videos among various ones. As a result, it is difficult for edge nodes to cache adequate user-desired contents due to limitations in cache

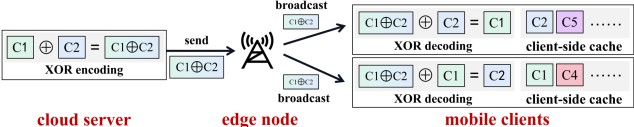

**Figure 1: File flows example for NCTM**

capability, leading to low caching efficiency [16]. According to Nokia's report [17], the hit rate of edge caching solutions is only 29% for short video delivery. Therefore, edge caching is not always as effective as expected.

Recent studies [18, 19] have revealed that using client-side caches within a peer-to-peer (P2P) network can significantly reduce bandwidth pressure caused by redundant video transmissions [20]. P2P is a promising solution to increase the cache hierarchy and can compensate for the limited edge node capability. However, the majority of short video deliveries are from mobile devices, e.g., 97% of TikTok video deliveries [6]. The expensive uploading data bills or the data cap make P2P not a feasible solution. Therefore, we aim to make better use of the client-side cache to compensate for the lack of edge cache capacity, while avoiding the upload traffic costs.

Coded cache [21–24] could provide a solution. It avoids transmission redundancy by transmitting coded files without incurring additional traffic charges. It is based on merging two separate video files into one coded file at the server and forward it to the destination. The most commonly used encoding approach is XOR-merging (Exclusive OR). After receiving the coded file, the destination device separates the original files by locally cached content prefetched during network idle time. So the coded cache technique requires fewer transmission files to transmit all data. But it also means some contents need to be cached in the destination device before use, and devices need to be active simultaneously to receive the coded files in time. However, in practice, short video applications generally do not allow network usage when they are closed, and the server lacks mechanisms to confirm the apps running status or interact with the users' cache. The above-mentioned problems hinder the practical deployment of the coded cache solution.

In this paper, we propose an innovative **Novel Coded Transmission Mechanism (NCTM)** for efficient short video delivery. With deep insight on the characteristics of short video delivery, NCTM avoids content prefetching or uploading through an effective cooperation between the cloud, edge, and mobile clients. Particularly, NCTM caches videos that the user **has already watched** for subsequent decoding rather than prefetching ones.

As shown in figure 1, the cloud server in NCTM applies XOR operations to combine video files into a single coded file, and sends it to the edge node (coded transmission). Then the edge node broadcasts the coded file to specific mobile clients. The mobile clients will decode the coded file by applying XOR operations with the assistance of previously watched and cached videos. To make sure that the coded file can be successfully decoded, we should divide the clients into multiple groups, ensuring that any two clients

within a group have previously watched and cached the files being requested currently by each other (group divided problem). So we introduce a `User Cache Table (UCT)` and a `Transmission Matching Graph (TMG)` that record the cache status and explore coded transmission opportunities. Recognizing the similarity between the above-mentioned requirement and the mathematical concepts of "cliques", we model the group divided problem as the clique cover problem [25, 26] and propose the novel *Minimum Clique Coverage algorithm*. This algorithm involves multiple linear-time searches to find a relatively optimal solution. Furthermore, we introduce the new *Recommendation Reorder algorithm*, which modifies the playback order by bringing forward videos scheduled for later to create more opportunities for coded transmission. Finally, we propose a *Client-side Cache Update method* based on the video recommended queue, which is informed by the recommendation system in short video services.

To evaluate the performance of NCTM, we utilized the kuaiRec dataset [27] and collected real user interactions with the Kuaishou app on August 12, 2020, totaling 117,977 records. For each client, edge node, and cloud server, we created separate docker containers [28] to simulate video requests and playback behaviors based on their app usage time. The results show that under sufficient bandwidth, NCTM reduces the average network load by 3.02%-14.75%, and cuts the peak traffic by 23.01%. With limited bandwidth, NCTM reduces 43%-85% of rebuffering events and increases video occupancy in the user's buffer by 1.21x-13.53x. Furthermore, we use the NCTM to assist the edge caching, proving that they are compatible. In addition, we demonstrate that NCTM can achieve real-time performance on the server and provide reference values for the hyperparameters in the proposed approach.

In summary, our contributions are as follows:

- We propose the innovative **NCTM** that sends XOR-coded files and utilizes the client-side cache to store watched videos for file decoding. This approach reduces the network load without data prefetching or uploading.
- We design the *Minimum Clique Coverage algorithm* and the *Recommendation Reorder algorithm* to find a relatively optimal solution for the client dividing problem with linear time complexity, which ensures successful decoding for the coded files in coded transmissions. The *Client-side Cache Update method* based on the video recommended queue is also proposed.
- We devise trace-driven experiments to emulate the NCTM and verify its effectiveness in the reduction of bandwidth consumption, buffer variation, and rebuffering frequency.

The remaining of the paper is structured as follows. In section 2, we summarize the NCTM related work, in section 3, we illustrate the overall structure of NCTM and in section 4, NCTM is described in details. In section 5, the experimental results are presented. Finally, we give a brief conclusion of our work in section 6.

## 2 RELATED WORKS

The popularity of short video applications leads to massive video traffic and introduces a significant load on CDNs.

**Edge caching** [29, 30] is a key approach to alleviate the CDN load. It utilizes cache-enabled edge servers, such as base stations and

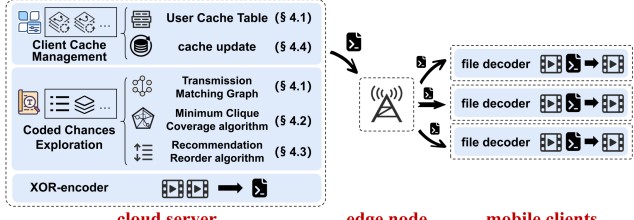

**Figure 2: Overall structure of NCTM**

smart gateways, to store popular contents, so that these contents can be transmitted directly from the caches instead of from the remote cloud [12]. As these cache edges are closer to the users, there is a reduction in the traffic load on the core network. The cache hit rate is an essential metric to evaluate the performance of edge caching. For short video deliveries, [31] proposes to consider the number of views and likes as the popularity basis to choose the edge cache content. Furthermore, [32] takes user preferences into consideration. To achieve a higher hit rate, [33] introduces a multi-agent deep reinforcement learning algorithm where each edge learns its own best policy.

**P2P-CDNs** [34–37] enables content caching on mobile phones by utilizing wireless channels for cache sharing, also known as P2P sharing. With a P2P approach, users can send their stored content to other users, effectively balancing the upstream and downstream transfers, and increasing the cache hierarchy [38]. [39] considers different network environments. High-quality Internet access shares its resources (e.g., bandwidth) with slower or unreliable ones. [40] takes advantage of user social information (friends/ followers), using the friend list to identify additional P2P sharing.

**Coded cache** [21–24] utilizes network coding techniques, which aggregate (encode), broadcast, and separate (decode) data messages in the cloud, edge node, and client devices, respectively. This technique results in lower traffic load for data transmission. For example, if user A caches video file $c_1$ and user B caches video file $c_2$, when user A requests video file $c_2$ and user B requests video file $c_1$, the server can transmit a single coded file $c_1 \oplus c_2$ (the symbol $\oplus$ represents bitwise XOR) instead of two separate files $c_1$ and $c_2$. It can deliver the coded file simultaneously to both users by broadcasting at the edge node, and users can decode it based on their local files (e.g., for user A: $c_1 \oplus c_2 \oplus c_1 = c_2$).

The previous works made important contributions in terms of edge caching, P2P-CDNs and coded cache solutions individually, but they have not combined them in an approach that reduces the load for short video deliveries as proposed by NCTM.

## 3 NCTM ARCHITECTURE AND PRINCIPLE

The overall NCTM architecture is shown in figure 2. The mobile clients cache their watched video files based on their cache capabilities, and the cloud server sends the coded files desired by multiple clients to the edge node. Then the edge node broadcasts the coded file to the mobile clients. These clients decode the coded file based on the cached files to obtain the desired one. To make sure the coded file can be successfully decoded, we need to ensure that all the files involved in the coded file have been cached at the clients, except the desired one. So the challenge lies in finding **how to efficiently and accurately identify the clients with the above**

**cache situation among multiple ones**. We employ the following designs to address this challenge.

The cloud server includes three key modules: **Client Cache Management**, **Coded Chances Exploration**, and **XOR-encoder**. The Client Cache Management informs the cache status of the client's devices. We design the User Cache Table (UCT) (section 4.1) to record the cached files in clients. Due to the limited cache capability, we specify the corresponding cache update policy (section 4.4). The Coded Chances Exploration module finds the coded transmission opportunities, thus minimizing bandwidth consumption. Since the condition of multivariate-coded transmission is similar to the concept of clique in mathematics, we design the Transmission Matching Graph (TMG) and reduce the chances exploration problem to the clique cover problem [25, 26] (section 4.1). To solve this NP-Hard problem, we designed the *Minimum Clique Coverage algorithm* to find a relative optimal solution within linear time complexity (section 4.2). Moreover, based on our observation, switching the playback order of short video can create more coded transmission opportunities, so we further proposed *Recommendation Reorder algorithm* (section 4.3). File encoder merges several origin video files into one coded file by XOR before transmission.

The edge nodes copy the coded file and broadcast it to clients. Clients decode the coded file and extract the original files, as desired.

## 4 CODED TRANSMISSION MECHANISM

### 4.1 Definitions

We assume that there are currently $N$ short video files, denoted as $\mathbb{C} = \{C_1, C_2, ..., C_N\}$. Short videos are commonly delivered with an adaptive bitrate paradigm [41, 42]. Now we assume that each short video consists of only one video chunk; this will be generalized to the common case in the next section. Suppose there are $K$ mobile clients, denoted as $\mathbb{U} = \{U_1, U_2, ..., U_K\}$. Each client is assigned an independent recommendation queue, storing short videos recommended for this user. We denote $v_{ij}$ as the $j$-th video pushed to the $i$-th user ($v_{ij} \in \mathbb{C}$ determined by the recommendation system). Therefore, for the user $U_i$, the recommendation queue can be denoted as $V_i = \{v_{i1}, v_{i2}, ...\}$. At this point, we define the UCT as $\mathbb{T} = \{t_{ij}\}$, where $t_{ij} = 1$ indicates that user $U_i$ has **watched and cached** video $C_j$, and vice versa ($i \in [1..K], j \in [1..N]$). Thus, for the user $U_i$, all the videos in their client-side cache can be recorded as $T_i = \{C_j | t_{ij} = 1, j \in [1..N]\}$.

For example, here we assume that user $U_a$ has cached video $C_j$, and user $U_b$ has cached video $C_i$. It can be denoted as $t_{aj} = 1$ and $t_{bi} = 1$, respectively. When the user $U_a$ needs the video $C_i$, and user $U_b$ needs video $C_j$, the coded file $C_i \oplus C_j$ can be transmitted. So the condition for coded transmission is that **they have cached the files needed by each other** (e.g., $t_{aj} = 1$, $t_{bi} = 1$), which is to ensure successful decoding in the user device (e.g., for user $U_a$, $(C_i \oplus C_j) \oplus C_j = C_i$, where $C_j$ is watched and cached video file of user $U_a$). This is defined as a binary-coded transmission.

Similarly, here we assume that there are three users $U_a, U_b$, and $U_c$ who need videos $C_i, C_j$, and $C_k$, respectively. When $t_{aj} = t_{ak} = 1$, $t_{bi} = t_{bk} = 1$ and $t_{ci} = t_{cj} = 1$, indicating that each user has cached the files needed by the other two users, the coded file $C_i \oplus C_j \oplus C_k$ can be transmitted. The client devices can also successfully decode the coded file with their cached video files. (e.g., for user $U_a$, decoding

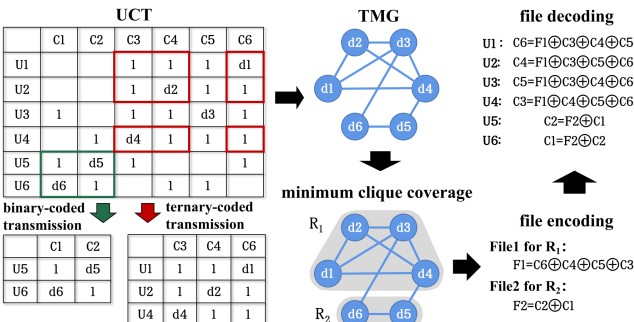

**Figure 3: Example of NCTM with 6 users and 6 videos**

can be achieved through $(C_i \oplus C_j \oplus C_k) \oplus C_j \oplus C_k = C_i$). This is defined as a ternary-coded transmission. Similar operations can be generalized to multivariate-coded transmission involving x video files ($x \geq 2$).

Clearly, a larger value of $x$ indicates merging more video files into one coded file, resulting in fewer number of transmissions and less bandwidth consumption. Therefore, we prefer the coded transmission that covers more users. To make sure the client devices can successfully decode, multivariate-coded transmission requires that any two users satisfy the condition for binary-coded transmission. This requirement can be associated with the mathematical concept of "cliques" (a clique is a complete graph where any two nodes are connected by an edge [43]). So we design the TMG as $\mathbb{G} = (E, D)$ to model the multivariate-coded transmission exploration problem as a graph theory problem. $D$ is the set of vertices representing the active users currently. If $d_i \in D$, it means the user $U_i$ is **watching videos**, and vice-versa. $E$ is the set of edges. If $e_{ij} \in E$, it means there is an edge between nodes $i$ and $j$, indicating that users $U_i$ and $U_j$ currently satisfy the **condition for binary-coded transmission**, and vice-versa.

Let us denote the watching video of user $U_i$ as $v_i^*$ ($v_i^* \in V_i$). In this case, for users $U_i$ and $U_j$, if $t_{iv_j^*} = t_{jv_i^*} = 1$, it indicates that they have cached the files desired by each other, binary-coded transmission can be achieved, i.e., $e_{ij} \in E$. Following this rule, we can construct the $\mathbb{G}$ with $K'$ nodes, where $K'$ is the number of active users ($K' \leq K$). In TMG, if we can find a clique with $x$ nodes, it means that among these $x$ users, any two users satisfy the condition for binary-coded transmission, thus satisfying the condition for multivariate coded transmission involving $x$ users. On the other hand, all user requests need to be fulfilled, so any node in current TMG needs to be covered by a clique (a single node is also considered a clique, we define it as the separate clique). Therefore, the problem can be transformed into finding the **minimum number of cliques** that **cover all nodes** in the TMG. We call this the group divided problem.

Formally, we denote $\mathbf{R} = \{R_1, R_2, ...\}$ as all cliques in TMG, where the i-th clique is represented as $R_i = \{d_x, d_y, ...\}$. We must ensure $R_1 \cup R_2 \cup ... = D$ and minimize $|\mathbf{R}|$. This problem is a classical clique cover problem and has been proven to be NP-hard when the degree of vertices is greater than 6 [25, 26]. Finding an optimal solution requires high complexity. In this paper, we propose the *Minimum Clique Coverage algorithm*, which utilizes the previous result and a single traversal to solve this problem with linear time complexity. The details of this algorithm are presented in section 4.2.

| request file | current client-side cache | | | | UCT | | | | |
|---|---|---|---|---|---|---|---|---|---|
| | T1 | T2 | T3 | T4 | | C3 | C4 | C5 | C6 |
| $v_1^*$ C6 | | | | | U1 | 1 | 1 | 1 | d1 |
| $v_2^*$ C4 | C3 | C5 | C4 | C4 | U2 | 1 | d2 | 1 | 1 |
| $v_3^*$ C5 | C4 | C6 | C6 | C6 | U3 | 1 | 1 | d3 | 1 |
| $v_4^*$ C3 | C5 | C3 | C3 | C5 | U4 | d4 | 1 | 1 | 1 |

| request time | 0s | 2s | 4s | 6s | 8s | 10s | 12s | 14s | 16s | 18s | 20s | 22s |
|---|---|---|---|---|---|---|---|---|---|---|---|---|
| U1: 4s-14s | | | $c_{61}$ | $c_{62}$ | $c_{63}$ | $c_{64}$ | $c_{65}$ | | | | | timeline |
| U2: 0s-22s | $c_{41}$ | $c_{42}$ | $c_{43}$ | $c_{44}$ | $c_{45}$ | $c_{46}$ | $c_{47}$ | $c_{48}$ | $c_{49}$ | $c_{4,10}$ | $c_{4,11}$ | |
| U3: 8s-22s | | | | | $c_{51}$ | $c_{52}$ | $c_{53}$ | $c_{54}$ | $c_{55}$ | $c_{56}$ | $c_{57}$ | |
| U4:18s-22s | | | | | | | | | | $c_{31}$ | $c_{32}$ | |

| TMGs | time 0-4s | time 4-8s | time 8-14s | time 14-16s | time 18-22s |
|---|---|---|---|---|---|

**Figure 4: An example of asynchronous user requests**

Figure 3 illustrates an example containing 6 users and 6 videos, and gives intuitive cases of binary-coded transmission and ternary-coded transmission. In the example, two cliques are used to cover the TMG, with $R_1 = \{d_1, d_2, d_3, d_4\}$ and $R_2 = \{d_5, d_6\}$. Therefore, in the coded transmission, two coded files need to be transmitted. $F_1$ merging 4 original video files $C_6, C_4, C_5, C_3$ and $F_2$ merging 2 video files $C_2, C_1$. The decoding process will be completed in the client devices with cached video files. In practical deployment, we group users with close geographic proximity and similar playback queues together to perform coded transmission. So the scale of the UCT and the TMG will not be very large. In our experiments, there are 1398 users and 10230 videos, and the storage capacity for the UCT is 202MB, which does not impose a significant load on the server.

## 4.2 Minimum Clique Coverage algorithm

To ensure smooth video playback, short video transmission uses the adaptive bitrate paradigm. We define a video $C_i$ as composed of multiple video chunks, denoted as $C_i = \{c_{i1}, c_{i2}, ...\}$. Note that the UCT and TMG are dynamically maintained as the videos play.

As shown in figure 4, users $U_1 - U_4$ have already watched and cached parts of the videos. For example, the user $U_1$ cached the videos $C_3, C_4, C_5$, i.e., $T_1 = \{C_3, C_4, C_5\}$. Now let us consider that user $U_1$ is watching video $C_6$, user $U_2$ is watching video $C_4$, i.e., $v_1^* = C_6$, $v_2^* = C_4$, and so on. In the previous example, we assumed that all users request videos strictly at the same time. However, in this one, the user request time is different, and there are variations in the duration of the videos. For example, user $U_1$ watches video $C_6$ within the time range of 4-14 seconds, user $U_2$ watches video $C_4$ within the time range of 0-22 seconds, and so on. Therefore, the TMG also changes within different time intervals. The figure depicts the TMGs corresponding to the current 5 intervals. At this point, encoding operations are performed between different video chunks rather than the entire video (for example, at the 8th second, $c_{63} \oplus c_{45} \oplus c_{51}$ is encoded for transmission). It is evident that the TMG changes only when a user either completes a video playback (e.g., 14th second) or starts watching a new video (e.g., 0th second, 4th second, 8th second, and 18th second). During each time interval, the TMG remains unchanged.

After completing a video playback, the user will leave the current TMG. Since the subgraph of a clique remains a clique, the removal of a node will not break the current clique. In the given example, at the 14th second, user $U_1$ (node $d_1$) leaves the TMG, but users $U_2$

(node $d_2$) and $U_3$ (node $d_3$) can still form a clique. When a user starts playing a new video, a new node is added to TMG, but the cliques formed by all the existing nodes remain unchanged. So we can decide whether the new node can be added to an existing clique (maintaining the same total number of cliques) or the new node forms a separate clique (increasing the total number of cliques by 1). A similar situation can be seen in the example at the 8th second when user $U_3$ joins the TMG as a new node $d_3$. According to the rules, node $d_3$ has edges with nodes $d_1$ and $d_2$, so it can be added to the clique formed by these two nodes, forming a clique with three nodes and achieving ternary-coded transmission.

Therefore, whenever a new request arrives, we need to add a new node to the TMG and try to incorporate this node into existing cliques. We can judge whether the incorporation condition is met by iterating through all existing cliques and checking whether all nodes can connect to the new node. If a suitable clique is found, the new node will be added to this clique, maintaining the existing number of cliques and the coded files to be transmitted unchanged. Otherwise, the new node forms a separate new clique, leading to an increase in both the number of cliques and the coded files to be transmitted.

However, the aforementioned approach often yields poor performance, as shown in figure 5. As new requests from user $U_5$ and $U_6$ arrive at time $t_2$ and $t_3$, new nodes $d_5$ and $d_6$ are added to current TMG. Since they do not have edges connecting to all nodes in the existing clique, they will form new cliques. Clearly, the optimal solution involves only two cliques, but the result at time $t_3$ contains three. To address this issue, we propose a search algorithm with linear time complexity, the *Minimum Cliques Coverage algorithm*, to find a relatively optimal solution based on the previous results. The algorithm includes $T$ iterations, where $T$ is a hyperparameter that controls the computational cost. In each iteration, the following steps are executed:

(1) Randomly select a node $d_i$ among all separate cliques. We assume that $d_i$ comes from the clique $R_x$.
(2) Randomly select an edge $e_{ij}$ from all the edges connected the node $d_i$. Obviously $e_{ij}$ connects to node $d_j$. We assume that $d_j$ comes from the clique $R_y$.
(3) Let $d_j$ exits the original clique $R_y$ and joins the clique $R_x$ where $d_i$ belongs to. The processed cliques are denoted as $R_y'$ and $R_x'$, respectively.
(4) Iterate through all the nodes ($d_k$) from $R_y'$. Check if there exist nodes that have edge connected to $d_i$ ($e_{ik} \in E$).
(5) Make these nodes ($d_k$) exit the original clique $R_y'$ and join the clique $R_x'$. The processed cliques are denoted as $R_y''$ and $R_x''$, respectively.
(6) Iterate through all the remaining separate cliques ($d_k$). Check if one can be added to the clique $R_y''$ after excluding some nodes.
(7) If such a separate clique exists, node $d_k$ joins the clique $R_y''$. The processed clique is denoted as $R_y'''$. Now a solution is found and the algorithm terminates. Otherwise, all adjustments are retained and the next iteration starts.

The pseudocode is shown in appendix A, and we prove this algorithm in appendix B. During the iteration, if such a separate

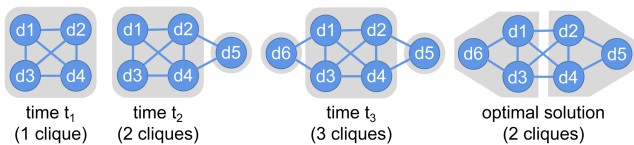

**Figure 5: Naive solutions often fail to achieve optimal solutions**

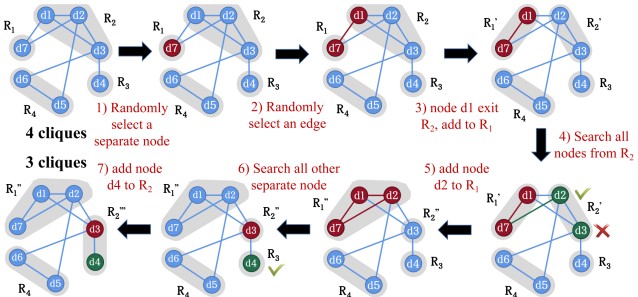

**Figure 6: An example of the Minimum Cliques Coverage algorithm**

clique is found in Step 7, it means that we successfully merge two separate cliques $(R_x, R_z)$ and one non-separate clique $(R_y)$ into two non-separate cliques $(R_x'', R_y''')$. The total number of cliques and the coded files to be transmitted reduces by one, and the algorithm terminates. Otherwise, we still need to preserve the above operations so that the next iteration can discover more matching opportunities. Figure 6 illustrates an example of the *Minimum Cliques Coverage algorithm*. Here the separate cliques $R_3(d_4)$ and $R_1(d_7)$ are merged with the clique $R_2$ containing three nodes $\{d_1, d_2, d_3\}$ to form the new clique $R_1''$ with three nodes $\{d_1, d_2, d_7\}$ and the new clique $R_2'''$ with two nodes $\{d_3, d_4\}$. At this point, the clique $R_3$ disappears.

This algorithm is executed whenever a new request arrives and the newly added node in the TMG cannot join an existing clique (i.e., forming a separate clique). In each round, all nodes are traversed at most three times, resulting in a computational complexity of $O(KT)$, where $K$ is the number of users in the current TMG, and $T$ is a hyperparameter that controls the number of iterations. We will further explore the time complexity of this algorithm and the impact of $T$ on the success rate of the search in our experiments.

The major benefit of the above algorithm is that we do not rely on any prior knowledge (such as video popularity or predictions of user viewing behavior). Instead, we dynamically adjust the cliques through the UCT and TMG. This allows us to determine which users currently satisfy the coded transmission conditions. Furthermore, with the assistance of the *Minimum Clique Coverage algorithm*, we utilize the historical results to explore a better solution for the clique cover problem in linear time complexity. This allows us to use fewer cliques, which means fewer coded files to be transmitted, thereby reducing bandwidth consumption.

### 4.3 Recommendation Reorder algorithm

Based on our observations, the opportunities for coded transmission are related to users currently watching videos. In some cases, playing a video located further in the recommended queue may create more chances for coded transmission than playing the very next video. A typical example is presented in figure 7.

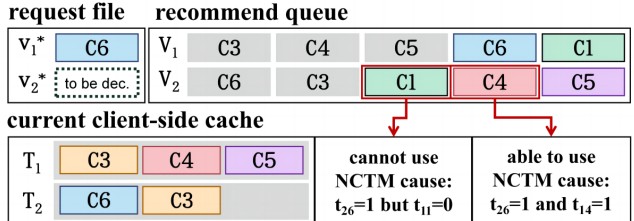

**Figure 7: Example for the Recommendation Reorder Algorithm**

The example contains two users, $U_1$ and $U_2$. User $U_1$ has a recommendation queue $V_1 = \{C_3, C_4, C_5, C_6, C_1, \ldots\}$, and videos $C_3$, $C_4$ and $C_5$ have been watched and cached in the client device, i.e., $T_1 = \{C_3, C_4, C_5\}$. User $U_1$ is currently watching the video $C_6$, i.e., $v_1^* = C_6$. Similarly, $V_2 = \{C_6, C_3, C_1, C_4, C_5, \ldots\}$, and $T_2 = \{C_6, C_3\}$. Since the previous video has just ended, no video is being played by user $U_2$. Now we find that if video $C_1$ is played next based on the recommendation queue, since $t_{11} = 0$, NCTM cannot be used between users $U_1$ and $U_2$. However, if video $C_4$ is chosen to be played next (also in the recommendation queue, but not the very next one), we have $t_{26} = t_{14} = 1$, and NCTM can be used. Specifically, if the user $U_2$ chooses video $C_1$ as the next one, there will be no edges connecting to node $d_2$ in the TMG, and the node $d_2$ will inevitably form a separate clique. However, if $U_2$ chooses video $C_4$, in the TMG, the node $d_2$ will have at least one edge. Even if it forms a separate clique, with the *Minimum Clique Coverage algorithm*, it also offers the potential for coded transmission.

For push-playback short videos, there are no content correlations between two consecutive videos. Therefore, changing the playback order of videos without altering their content will not significantly affect the user experience. The above example illustrates that in certain situations, reordering the recommendation queue can create more opportunities for coded transmission. Given that the users might end, skip, or re-watch videos at any time, this makes it difficult to predict users' viewing behaviors. Therefore, we propose the *Recommendation Reorder algorithm*, which focuses on short-term benefits. Whenever a user switches videos, this algorithm will filter the videos that can trigger **NCTM** based on the current UCT and select one based on the recommendation queue. This method does not change the recommendation queue results, but changes the video playback order by prioritizing the ones that enable coded transmission. Thus, it creates more chances for coded transmission. The pseudocode is shown in appendix D. The algorithm considers the next $G$ videos in the recommendation queue of user $U_i$. For each video, it iterates through all users to determine if the binary-coded transmission can be used. The algorithm can be described as the following process:

(1) Traverse the next $G$ videos in $V_i$, denote as $C_j$.
(2) For each $C_j$, traverse all other active users, denote as $U_k$. Check if one satisfies the requirement of binary-coded transmission, i.e., $t_{kj} = t_{iv_k^*} = 1$.
(3) If such a user exists, let $v_i^* = C_j$. That is, the next video to be played is $C_j$. Otherwise, still play the first video in $V_i$.

Here $G$ is a hyperparameter that decides the maximum number of look-forward videos as well as limits the computational cost.

This algorithm involves two iterations. So the time complexity is referred to $O(GK)$, where $G$ represents the number of next videos considered in the recommendation queue, and $K$ represents the number of users watching videos. To further prune the search path, we can avoid the case where $t_{iv_j^*} = 0$ during the first traversal, i.e., excluding the users who are watching the video not cached by user $U_i$. That is because the user $U_i$ can not meet the coded transmission condition whatever the next video is, and it is impossible to change others playing videos.

Currently, most short video applications pre-cache the first video chunk of several subsequent videos in the recommendation queue, to alleviate the stalling and rebuffering issues caused by rapid video switching. For example, TikTok pre-caches the first video chunk of the next five videos [44]. Assuming the application pre-caches the first video chunk of $P$ subsequent videos, it should ensure $G \leq P$ to mitigate the risk of increased stalling and rebuffering events.

### 4.4 Client-side cache update method

As discussed in Section 1, both edge caching and client-side cache have limited storage capabilities. Compared to edge caching, user devices (such as smartphones) may be more restricted. In this section, we will discuss how to perform cache placement/replacement to better cooperate with the *Minimum Clique Coverage algorithm* and *Recommendation Reorder algorithm*.

We define the maximum storage capacity for user $U_i$ as $M_i$, thus we need to ensure that $\sum_{C_j \in T_i} |C_j| \leq M_i$, where $T_i$ is the cached videos of user $U_i$ and $|C_j|$ represents the file size of video $C_j$. Suppose that after watching video $C_k$, $(\sum_{C_j \in T_i} |C_j|) + |C_k| > M_i$, indicating that the user $U_i$ cannot store all watched videos on the user device. Therefore, it is necessary to determine which content should be replaced.

For video $C_k$, we cache video files so we can decode the coded files with the assistance of them in the future. Hence, cached videos $C_k$ are only valuable if they are watched by other users later. Using the "push playback" style, a recommendation system can keep track of the video playback list. Therefore, we can estimate the earliest possible time when the video $C_k$ will be used for file decoding. As NCTM mainly focuses on short-term benefits, we prefer the files that can create coded transmission opportunities in the short term and replace those that would take longer to be used.

Formally, we define $F(C_k)$ as the earliest occurrence time of video file $C_k$ in the recommendation queue. That is, $F(C_k) = \min(j)$ s.t. $v_{ij} = C_k$ for all $i \in [1..K]$, where we find the smallest $j$ among all $K$ users' recommendation queues such that $v_{ij} = C_k$. This is the earliest possible time when the current video could be used for decoding. For the user $U_i$, we can sort the videos in $T_i$ based on $F(C_k)$ and replace the videos that would take longer to be used.

Note that from the perspective of the *Recommendation Reorder algorithm*, it can be regarded as $F^*(C_k) = \max\{1, F(C_k) - G\}$, where $G$ is the hyperparameter defined in the algorithm description. This is because the *Recommendation Reorder algorithm* can look ahead $G$ videos based on the current recommendation queue, thus this video could potentially be played ahead by up to $G$ videos. Additionally, when we consider edge cache, if video $C_k$ is cached in the edge cache, it means that requests involving video $c_k$ will not use NCTM and can be regarded as $F(C_k) = +\infty$.

### 4.5 Further considerations

According to our further observations, due to the spatial and temporal densification of of short videos, NCTM exhibits significant potential in practical applications. For the uncompleted watching events, we will consider weighted *TMG* in our subsequent work to explore more stable coded transmission opportunities. Furthermore, short video playback on mobile devices also exhibits a significant degree of randomness. For instance, users may slide the progress bar to skip some uninteresting scenes, network conditions can affect the video bitrate, and user movements can impact the connection status of edge nodes. We have taken these issues into consideration and discussed them in detail in appendix C.

## 5 EVALUATION

### 5.1 Methodology

We used the **traditional CDN delivery** and **edge cache** (size of 500MB with a LRU update method) approaches for comparison in order to evaluate the performance of NCTM. To replicate the users video-watching behavior, we utilized the kuaiRec dataset [27] and collected 117,977 real request records from 1,398 users on the Kuaishou app on August 6, 2020, involving 10,230 short videos. To simulate the network conditions, we used the MAWI [45] and FCC18 [46] network traces from August 12, 2020. In the actual implementation, we created separate Docker [28] containers for the server, the edge node, and each user device. Users sent requests to servers via edge nodes according to the real request records mentioned above, in order to replicate the video transmission and playback behaviors. Additionally, we used the Mahimahi network tool [47] to reply the network trace, aiming to closely reproduce the network conditions at that time. In the experiments, we considered two fundamental scenarios: the sufficient bandwidth network and the limited bandwidth network. For the former one, we mainly considered the network bandwidth utilization. For the latter one, we mainly focused on buffer variation and rebuffer events. More detailed settings are shown in appendix F.

### 5.2 Performance

**Under sufficient bandwidth:** Figure 8 presents the network load variations under high-speed connections (200Mbps) across CDN and edge caching approaches with and without NCTM, as well as a histogram of short video requests distribution. From the results, it can be observed that network load variations are closely related to the request frequency. The Pearson correlation coefficient between network load and request frequency is 0.77. The peak value of requests occurs at around the 29700th second, approximately at 8:15 AM. During this time, there are 118 requests within one minute (with a total of 1398 users), and the network load also reached its peak value. In CDNs delivery mode, the real-time throughput was 69.94 Mb/s without NCTM, while it is 53.84 Mb/s with the assistance of NCTM, cutting peak traffic by 23.01%. Similar results are observed in the edge caching mode, indicating that NCTM can effectively alleviate peak throughput pressure. Figure 9 presents the cumulative distribution function (CDF) of the normalized bandwidth usage. From the results, it can be observed that in CDNs delivery mode, the average network load decreased by 14.06% with NCTM versus

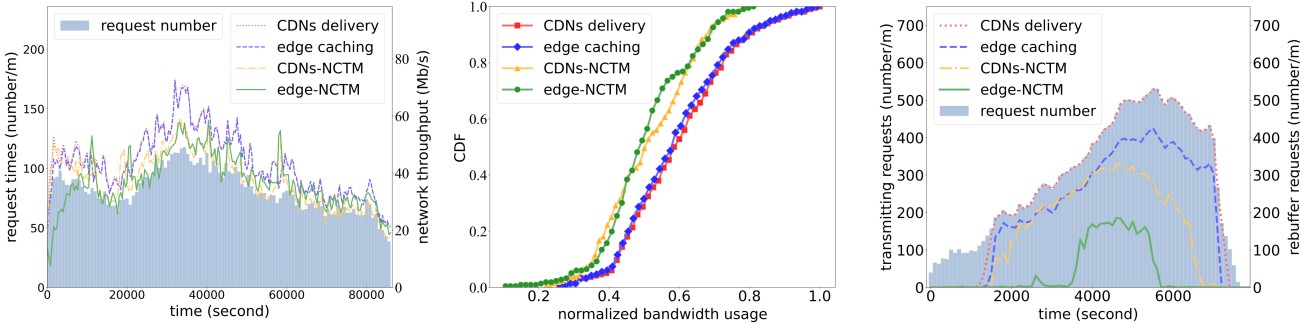

**Figure 8: The network throughput variation in different delivery modes under sufficient bandwidth**

**Figure 9: Normalized bandwidth usage in different delivery modes under sufficient bandwidth**

**Figure 10: The distribution of rebuffer events in different delivery modes under limited bandwidth**

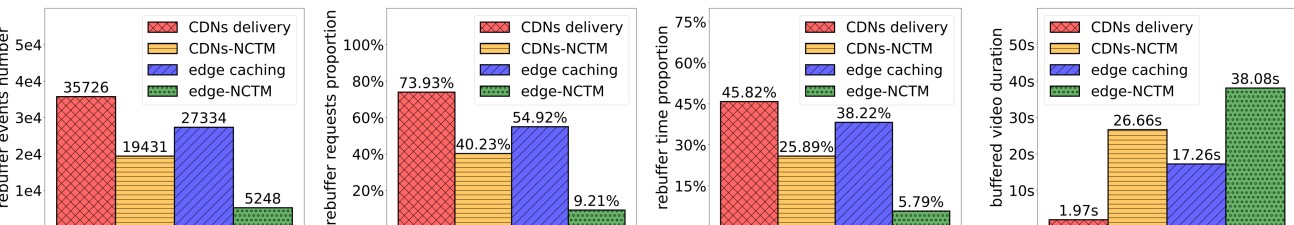

**Figure 11: The statistics of rebuffer events in different delivery modes under limited bandwidth**

**Figure 12: Proportion of rebuffered request in different delivery modes under limited bandwidth**

**Figure 13: Average proportion of rebuffer time in different delivery modes under limited bandwidth**

**Figure 14: Average buffered video duration in different delivery modes under limited bandwidth**

no NCTM. In the edge caching mode (with an edge cache size of 500MB), NCTM decreased network load by 13.30% compared to no NCTM. As a result of the change in the order of video playback, there may be a short period of time when NCTM's throughput exceeds that of baselines. From a global perspective, the NCTM is clearly superior to the baseline.

**Limited bandwidth scenario:** Figure 10 shows the distribution of rebuffer events using NCTM in both CDNs delivery mode and the edge caching mode in a limited bandwidth scenario (60Mbps). Figure 11 and 12 show the statistics of rebuffer events and the proportion of rebuffered requests respectively. From the figures, it can be observed that due to the limited bandwidth, in the CDN delivery mode, 73.93% of user requests suffer from rebuffer events. During peak request times, almost all users experience rebuffer events, resulting in a total of 35,726 rebuffer instances. NCTM relieves some of the bandwidth pressure, resulting in a 40.23% reduction in rebuffer events for user requests. This represents a decrease of 33.7% compared to the baseline. Total rebuffer instances decreased by 45.6% to 19,431. Similarly, in the edge caching mode, although rebuffer events have also decreased compared to CDNs delivery, the improvement is limited to users who request popular files. However, with the efficient cooperation of NCTM and edge caching, the number of user requests experiencing rebuffer events has been reduced to only 9.21%, and the total count of rebuffer events has decreased by 80.8% compared to the baseline.

Figure 13 shows the average proportion of rebuffer time across CDN and edge caching approaches with and without NCTM. Figure 14 shows the average buffered video duration during video playback

in client devices. It can be concluded that, due to frequent rebuffer events in the CDN delivery mode, the average buffered video duration is only 1.97 seconds. About 45.82% time of video watching is waiting for rebuffering. With NCTM, the average buffered video duration increases by 13.53x to 26.66 seconds, mitigating the effects of fluctuations in network performance. Similarly, in the edge caching mode, the average buffered video duration increases by 1.21x, and the proportion of buffer time decreases to only 5.79%. This optimization significantly improves user experience in limited network bandwidth conditions.

**Dynamic bandwidth scenario:** Figure 15 illustrates the average bandwidth usage in different delivery modes in a scenario where the network bandwidth ranges from 60Mbps to 100Mbps. Results reveal that when the bandwidth is relatively abundant (100Mbps 80Mbps), NCTM reduces the bandwidth usage rate by 3.02%-14.75%. Even in CDNs delivery mode, using NCTM outperforms edge caching mode. Under the constrained bandwidth (70Mbps), NCTM significantly alleviates the bandwidth pressure, reducing the time of full bandwidth occupancy by 15.28%, thereby substantially reducing rebuffering events.

**NCTM assisting edge caching:** To further explore the cooperation between NCTM and edge caching, we analyzed the number of requests in different transmission stages, including the number of hit-edge-cache requests and the number of NCTM requests, as shown in Figure 18. Result indicates that in the early stages, when the response workload of the edge cache is low, LRU can satisfy over 20% of the requests. However, with the video files increasing in the later stages, the edge cache becomes overwhelmed. Due to

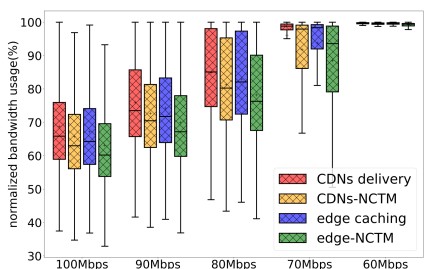

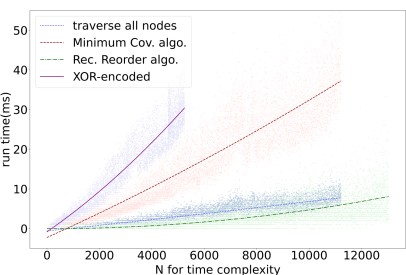

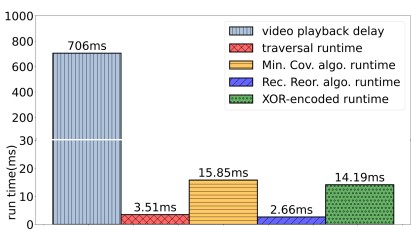

**Figure 15: Average bandwidth usage under different bandwidth (60Mbps-100Mbps)**

**Figure 16: Time complexity of NCTM-related algorithm**

**Figure 17: Average execution time of NCTM-related algorithm and video request delay**

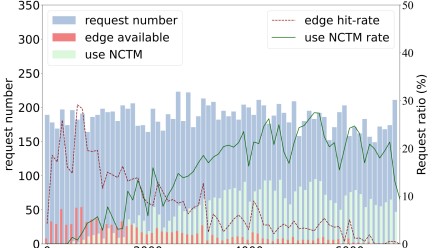

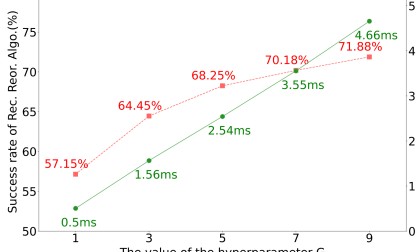

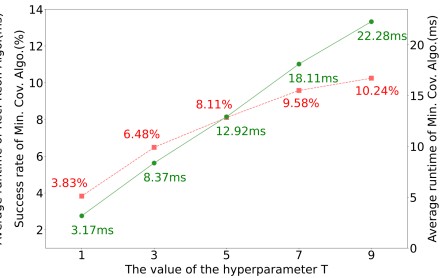

**Figure 18: Cooperation between NCTM and edge caching**

**Figure 19: Average execution time and success rate in different $T$**

**Figure 20: Average execution time and success rate in different $G$**

limited cache accumulation on the client side, NCTM does not occur frequently in the early stages. But in the later stages, as user caches accumulate, a large number of requests (over 25%) can use NCTM to transfer coded files. Therefore, edge caching and NCTM are not only compatible, but they are complementary and further reduce the network load by cooperation.

## 5.3 Time complexity and hyperparameters

**Time complexity:** We design an experiment to compute the time complexity of NCTM in order to assess NCTM's capability to process the received requests in real-time. Figure 16 illustrates the time complexity of NCTM. Figure 17 illustrates the relationship between the average execution time of NCTM and short video playback latency. The execution time of NCTM primarily includes four distinct components., i.e., cliques traversal, the *Minimum Clique Coverage algorithm*, the *Recommendation Reorder algorithm*, and the XOR encoding. Figure 16 shows four fitting curves, illustrating the actual execution time of the algorithms with different entity counts ($N$ for time complexity), such as the number of current nodes or cliques. From the results, it can be concluded that both the *Minimum Clique Coverage algorithm* and the XOR-encoding process show nearly linear growth, consistent with our analysis of time complexity in $O(N)$ (when $T = G = 5$). Although the *Recommendation Reorder algorithm* exhibits a slightly super-linear growth trend, its execution time remains below 10$ms$ even when $N > 10000$. Besides, the average execution time from opening the APP to the start of the first video playback is 706ms (tested with the Chrome browser for Tiktok). Considering that NCTM's execution time is much shorter than the video playback delay, we can conclude that NCTM can process requests in real time for short videos.

**Hyperparameter analysis:** In the *Minimum Clique Coverage algorithm* and the *Recommendation Reorder algorithm*, we set hyperparameters $T$ and $G$ to control the computational cost, respectively. To determine the values of $T$ and $G$, we design experiments with values $\{1, 3, 5, 7, 9\}$ and calculated the success rate for finding solutions and execution time under each parameter setting. Results in Figure 19 and figure 20 reveal that both $T$ and $G$ exhibit a nearly linear increase in average execution time as their values increase. When $T = 5$, the probability of reducing the number of cliques in the *Minimum Clique Coverage algorithm* is 8.11%, with an average execution time of 12.92$ms$; when $G = 5$, the probability of finding videos in the *Recommendation Reorder algorithm* is 68.25%, with an average execution time of 2.54$ms$. As $T$ and $G$ further increase, the success rate of finding solutions approaches a plateau, indicating that $T = 5$ and $G = 5$ are relatively optimal values. In practical applications, the parameters can be adjusted according to the server's computing resources.

## 6 CONCLUSIONS

As short videos become increasingly common, they put a significant strain on network resources due to the massive amount of data they transmit. Due to the time- and space-intensive nature of short videos, CDNs are faced with considerable bandwidth challenges. Moreover, edge caching and other solutions rely too heavily on predicting popular files. In this paper, we propose NCTM, which employs XOR-encoded files to make effective use of user-side caches, reducing bandwidth consumption and improving transmission efficiency. Trace-driven experiments show that compared to CDNs and edge caching, NCTM reduces bandwidth consumption, rebuffering events, and the reliance on caching popular files, ultimately improving the user experience when watching short videos.

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

# APPENDICES

# A  PSEUDOCODE FOR MINIMUM CLIQUE COVERAGE ALGORITHM

**Algorithm 1** Minimum Clique Coverage algorithm

**Input:** $\mathbb{G} = (E, D)$, $\mathbb{R}$, $T$
**Output:** $\mathbb{R}'$

1: **function** MAIN
2:     $\mathbb{R}' = \mathbb{R}$
3:     **repeat**
4:         randomly select $d_i \in D$, s.t. $d_i \in R_x$, $|R_x| = 1$
5:         randomly select $e_{ij} \in E$, s.t. $d_j \in R_y$, $|R_y| \neq 1$
6:         $R'_x = R_x \cup \{d_j\}$
7:         $R'_y = R_y - \{d_j\}$
8:         **for** $d_k \in R'_y$ **do**
9:             **if** $e_{ik} \in E$ **then**
10:                 $R''_x = R'_x \cup \{d_k\}$
11:                 $R''_y = R'_y - \{d_k\}$
12:             **end if**
13:         **end for**
14:         **for** $d_k$ s.t. $d_k \in R_z$, $|R_z| = 1$ **do**
15:             **if** $\nexists d_l$ s.t. $d_l \in R''_y, e_{kl} \notin E$ **then**
16:                 $R'''_y = R''_y \cup \{d_k\}$
17:                 $\mathbb{R}' = \mathbb{R}' - \{R_x, R_y, R_z\}$
18:                 $\mathbb{R}' = \mathbb{R}' \cup \{R''_x, R'''_y\}$
19:                 **return** $\mathbb{R}'$
20:             **end if**
21:         **end for**
22:         $\mathbb{R}' = \mathbb{R}' - \{R_x, R_y\}$
23:         $\mathbb{R}' = \mathbb{R}' \cup \{R''_x, R''_y\}$
24:         $T = T - 1$
25:     **until** $T = 0$
26: **end function**

# B  PROOF OF THE MINIMUM CLIQUE COVERAGE ALGORITHM

In the *Minimum Clique Coverage algorithm*, we randomly select a separate clique in each iteration and search for a solution to reduce the clique number. It's worth noting that although there are situations where the clique number remains unchanged, we still retain all adjustments made during this iteration. Figure 21 explains why we should retain the states. With simple TMG adjustments, larger cliques can be split into smaller ones, creating more opportunities for successful matches. In this example, the clique $R_1 = \{d_1, d_2, d_3, d_4\}$ is split into smaller clique $R'_1 = \{d_2, d_3, d_4\}$ in the first iteration, leading to a successful match in the next iteration. More generally, if we can find several **separate nodes** whose connected edges completely **cover all nodes** of a **non-separate clique**, we can then divide this non-separate clique into several parts putting in the separate cliques.

Formally, we define $P_i$ as the set of pointed nodes for all edges connected to node $d_i$. For example, in the first subgraph of figure 21, $P_1 = \{d_2, d_3, d_4, d_6, d_8\}$, which means node $d_1$ is connected to nodes $d_2$, $d_3$, $d_4$, $d_6$, and $d_8$. We also define $Q_i$ as the set of

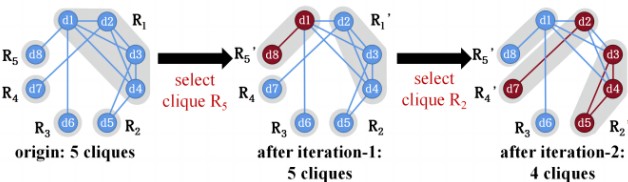

**Figure 21: An example of finding the solution through multi-step iteration search**

nodes belonging to the same clique as node $d_i$. For example, in the first subgraph of figure 21, we have $Q_1 = R_1 = \{d_1, d_2, d_3, d_4\}$, $Q_2 = R_1 = \{d_1, d_2, d_3, d_4\}$, and $Q_8 = R_5 = d_8$. If we can find several separate cliques (nodes) $\{d_i\}, \{d_j\}, \{d_k\}, ...$ and a non-separate clique $Q_x$ such that $Q_x \subseteq P_i \cup P_j \cup P_k \cup ...$, it means the clique $Q_x$ can be divided. For example, in the first subgraph of figure 21, we have $P_5 = \{d_3, d_4\}$, $P_7 = \{d_2\}$, $P_8 = \{d_1\}$, and thus $P_5 \cup P_7 \cup P_8 = \{d_1, d_2, d_3, d_4\}$. At the same time, we have $Q_1 = \{d_1, d_2, d_3, d_4\}$, so we have $Q_1 \subseteq P_5 \cup P_7 \cup P_8$. In this case, we can reconstruct the cliques $Q_1(R_1), Q_5(R_2), Q_7(R_4), Q_8(R_5)$ to form cliques $Q_5(R'_2) = \{d_3, d_4, d_5\}$, $Q_7(R'_4) = \{d_2, d_7\}$, $Q_8(R'_5) = \{d_1, d_8\}$ in the third subgraph. It means we form three non-separate cliques from three separate cliques and one non-separate clique.

This problem is a classic Set Cover Problem (SCP) that involves selecting specific sets and taking their union to cover all elements of a given set [48]. SCP is NP-hard in the strong sense, proven by Garey and Johnson[49]. In the *Minimum Clique Cover algorithm*, we utilize the characteristics of separate cliques. Each time, we select a separate clique and a non-separate clique, removing **all** nodes that are connected to the separate clique in the non-separate clique. Formally, it can be proven that if $Q_x \subseteq P_i \cup P_j \cup P_k \cup ...$, then for any set $P'$, we have $(Q_x - P') \subseteq ((P_i \cup P_j \cup P_k \cup ...) - P')$. Assuming it is for separate node $d_i$, we have $(Q_x - P_i) \subseteq ((P_i \cup P_j \cup P_k \cup ...) - P_i) \subseteq P_j \cup P_k \cup ...$. In other words, in the *Minimum Clique Coverage algorithm*, the SCP problem is broken into multiple subproblems which can be solved linearly. In each search, we use a separate clique to grab parts of a non-separate clique. This process continues until a suitable solution is found or the iteration limit has been reached. In the meanwhile,

Through the above example, we demonstrate that the *Minimum Clique Coverage algorithm* helps us reach the optimal solution without deteriorating the current situation. Therefore, we should retain all adjustments made during each iteration.

# C  FURTHER CONSIDERATIONS IN DETAILS

**The potential of NCTM:** NCTM requires users to sequentially watch the same videos and access the network through the same edge node (base station) synchronously. This may be unattainable in traditional on-demand video streaming. However, based on our deep insight on recent popular short video services, the spatial and temporal densification of short video delivery can meet this requirement. For a specific video, the number of viewers in a certain province or city can reach 40%, and over 70% of playback occurs within 48 hours after publishing. Videos with a clear regional bias exhibit even more pronounced spatial densification, making NCTM a more desirable choice for significant performance. (For more details, please refer to appendix E).

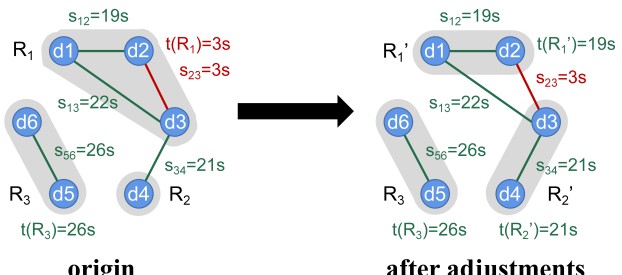

**Figure 22: Minimum Clique Coverage algorithm with weight edges in TMG**

**Uncompleted watching events:** An uncompleted watching event occurs when the user switches to the next video before finishing the last one. This is very common in short video playbacks. Obviously, the video chunks that are not downloaded before switches will not be saved in the client-side cache. Therefore, in the NCTM, if these video chunks are involved in the coded file, the client can not decode it successfully. We can make the node exit the current clique in TMG, forming a separate clique, this will evidently impair the performance of NCTM (the number of clique increase). A compromise solution approach is to assign a weight $s_{ij}$ for each edge in the TMG, representing how long the edge between node $d_i$ and $d_j$ can be maintained (how long the coded transmission can be sustained). Next, we denote $t(R_i)$ to represent how long the clique $R_i$ can be sustained. In the *Minimum Clique Coverage algorithm*, the problem is transformed into a weighted search problem. Greedy algorithms or heuristic search methods can be used to determine whether to retain the adjustments in each iteration. Figure 22 illustrates the process of the weighted algorithm. This iteration does not result in a decrease in clique number, but it extends their duration significantly. This implies a longer duration for coded transmission.

**Sliding the progress bar:** Sliding the progress bar indicates that the user may skip some uninterested content within one video. This is often accompanied by uncompleted watching events. In the NCTM, when a user slides the video, the edges of the corresponding node in the TMG need to be recalculated (sliding may cause some edges to disappear due to uncompleted watching). The node exits the current clique and then re-runs the *Minimum Clique Cover algorithm*. Furthermore, since NCTM is designed around continuous video playback, if there are missing video chunks in a video, it's recommended to only store the continuous video chunks from the beginning up to the missing portion in the client-side cache. The subsequent video chunks will be rarely used in file decoding.

**Video chunks with different bitrates:** Short videos are usually encoded as video chunks with different bitrates using the adaptive bitrate paradigm. In the NCTM, we treat video chunks with different bitrates as distinct videos. Fortunately, most short video platforms encode videos in only a limited range of bitrates. For instance, TikTok has only three bitrate options [44]. Furthermore, recently researched layered coding schemes [50, 51] are more compatible with NCTM. In these schemes, video files of different bit rates are compatible. High-bitrate files can also be used for decoding the coded files involving low-bit rate files.

**User movement:** The user rapid movement implies frequent changes to the connecting base stations, while the user's cache

remains the same. In NCTM, due to the involvement of edge nodes (base stations), changing base stations means that the existing coded transmission conditions cease to be satisfied. Any change can be regarded as exiting the original TMG and joining the new TMG with a new identity to explore coded transmission opportunities. However, NCTM is not suitable for situations with frequent handovers between base stations.

## D PSEUDOCODE FOR RECOMMENDATION REORDER ALGORITHM

**Algorithm 2** Recommendation Reorder algorithm

**Input:** $V_i$, $\mathbb{T}$, $G$
**Output:** $v_i^*$
1: **function** MAIN
2:     $flag = FALSE$
3:     **for** $C_j$ in $V_i$ next $G$ videos **do**
4:         **for** $U_k \in \mathbb{U}$ **do**
5:             **if** $t_{kj} = 1$ and $t_{iv_k^*} = 1$ **then**
6:                 $v_i^* = C_j$
7:                 $flag = TRUE$
8:             **end if**
9:         **end for**
10:         **if** $flag = TRUE$ **then**
11:             break
12:         **end if**
13:     **end for**
14:     **if** $flag = FALSE$ **then**
15:         $v_i^* = V_i$ next video
16:     **end if**
17:     **return** $v_i^*$
18: **end function**

## E THE POTENTIAL OF NCTM

NCTM is a mechanism designed for short video delivery. As the transmission and encoding/decoding of coded files require deep cooperation between edge nodes and multiple users, in practice, the users participating in a certain coded transmission should be within the coverage of the same base station. Additionally, they should have similar recommendation queues for short videos and watch them simultaneously. Our analysis of the characteristics of short video playback reveals significant spatial and temporal densification during playback. For a specific video, the number of viewers in a certain province or city can reach 40%, and over 70% of the playback occurs within 12 hours after publishing for some videos.

Based on the data collected from the Tiktok App, we have confirmed that there are significant regional differences in the viewership of videos across different categories. Videos with regional tendencies tend to concentrate their dissemination within specific provinces and cities. Furthermore, the playback time of videos is also concentrated within a short period of time after their release.

Due to the challenge of obtaining user viewing behaviors directly, we utilize the user comment behaviors as a substitute for exploration. Through packet capturing tools like Fiddler [52], we capture data packets related to video comments and extract information such as IP location and timestamps. This allows us to obtain

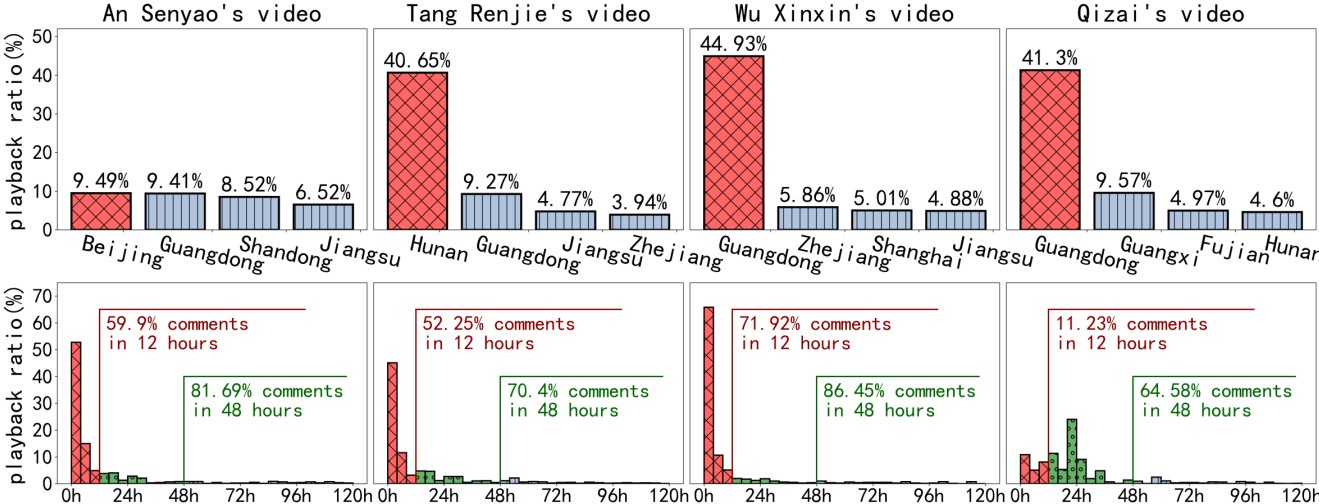

**Figure 23: spatial and temporal distribution of four typical influencers**

the distribution of comment IP locations and comment timestamps, reflecting when and where the user watch this video.

Concretely, we captured 100 videos from 25 Tiktok influencers. Four representative results are shown in figure 23. These four influencers are science popularizer An Senyao, food explorer Tang Renjie, real estate agent Wu Xinxin, and life enthusiast Qi Zai, with fan counts of 5.061 million, 12.963 million, 0.301 million, and 0.572 million, respectively. An Senyao's video content belongs to the popularization of science, discussing the correlation between ancient Europe and ancient China. The content with no regional bias, so the geographical distribution is quite balanced. The most concentrated place of commentators (from Beijing) accounts for only 9.49%. Tang Renjie's video content belongs to daily life, featuring a restaurant exploration in Changsha, Hunan province. Due to the specific regional information, commentators from Hunan province account for 40.65%. Wu Xinxin's video content belongs to finance, analyzing China's real estate transaction data in May 2023. However, she uses Cantonese, a dialect mainly spoken in southern China, indicating a clear regional bias. As a result, 44.93% of the commentators are from Guangdong province (the primary region where Cantonese is spoken). Qi Zai's video content also belongs to daily life, sharing daily cooking skills. But the video scenes are located in Guangzhou, Guangdong, indicating a certain regional bias. As a result, 41.3% of the users participating in the comments are from Guangdong.

On the other hand, figure 23 also illustrates the histogram of comment timestamp distribution. From the graph, it can be observed that a significant number of comments were created within 12 hours of the video being posted. The highest one (Wuxinxin's video) reached 71.92%. While some videos may become popular later on, they generally reach their peak of popularity within 48 hours. For instance, only 11.23% of the comments are made within the first 12 hours on Qi Zai's video, but it comes to 64.58% at the 48th hour. Furthermore, video content also affects their popularity. For instance, Wu Xinxin's video focuses on the real estate transaction data in May 2023, which possesses strong timeliness, showing stronger time densification. Contrarily, Qi Zai's videos revolve

around life skills, which do not emphasize timeliness, resulting in a longer duration of the popularity cycle.

Based on the results, spatial densification is mainly observed in videos that exhibit significant regional bias. There is a large number of video playbacks in a particular area. Videos with strong timeliness display temporal densification. The vast majority of the playback occurs within 48 hours of the video release. These create excellent usage scenarios for NCTM. In practice, we should pay more attention to the videos showing spatial and temporal densification. More frequent video playback implies a greater number of requests available for coded chances exploration which brings more potential improvements to NCTM.

## F OVERALL SETTING OF THE EXPERIMENT

**Comparison baselines:** We will compare NCTM with the following approaches. 1) *Traditional CDN delivery*: CDNs maintain all videos on the cloud servers. Mobile clients send requests to CDNs through base stations and the cloud servers will transmit the desired content to the mobile clients through the same path. 2) *Edge caching*: The edge nodes (e.g., base stations) cache popular files and use the Least Recently Used (LRU) method to update the content. When a client requests a file cached at the edge node, the edge node will prioritize providing it.

**Entity settings:** The video distribution over mobile networks generally involves three entities, i.e., content distribution servers (cloud servers), base stations (edge nodes), and client devices. After the user opens the app, the server sends a video list generated by the recommendation algorithm to the client. The client device then requests the videos through the base stations. If the video is available in the edge cache (located in the base station), it will be directly sent to the user. Otherwise, the cloud server will send the content to the base station, which will forward it to the user. In the evaluation, we use Docker [28] to simulate the above transmission process. At the beginning of the experiment, we create one Docker for the cloud server and another Docker for the edge node, separately running codes implemented with Python, to serve the clients.

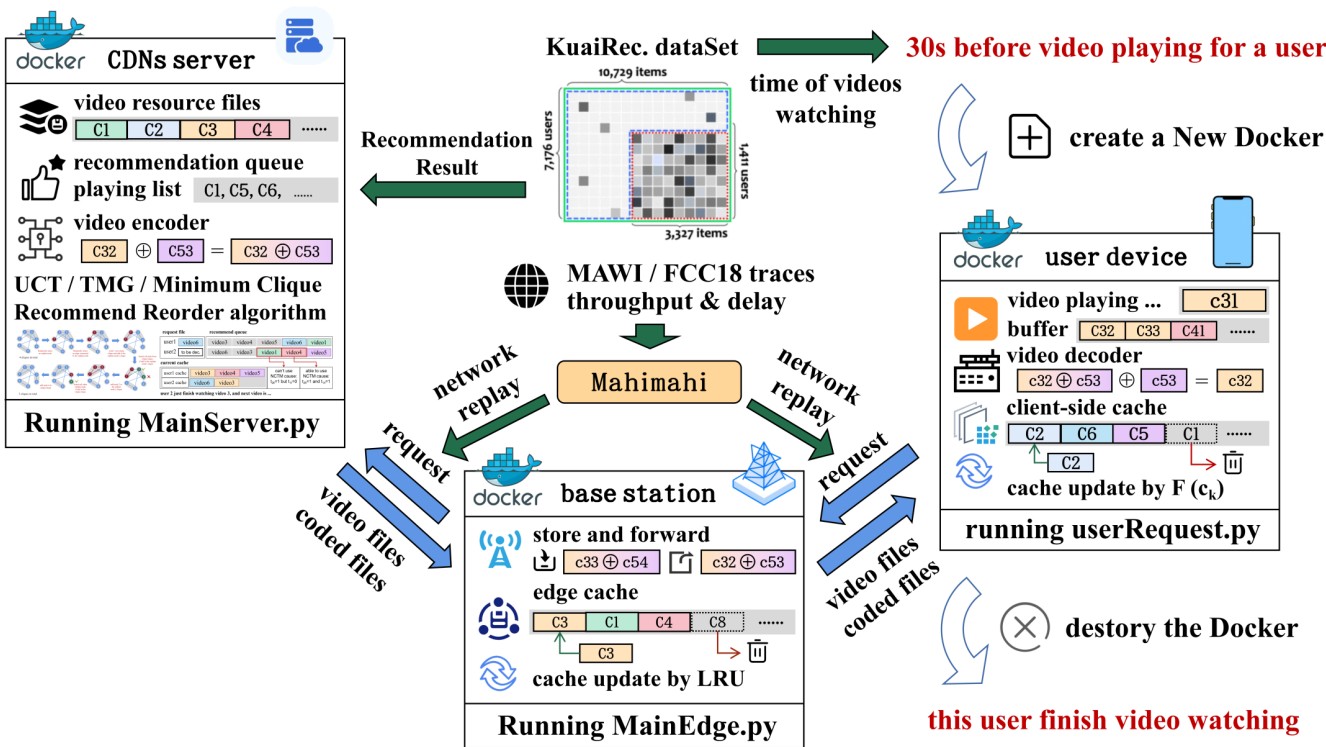

**Figure 24: Experimental Overall Structure Design**

For each client, a new Docker is created during the first 30 seconds before the user starts watching videos, and then simulating the behaviors of requesting video content as well as video playbacks. Once the client completes the session, the corresponding Docker will be deleted. Each Docker has its own port ID to send requests and transmit files via a TCP connection, with bandwidth variations simulated using Mahimahi [47]. The overall system is presented in figure 24.

**Viewing behaviors:** The experiment uses the kuaiRec dataset [27], which is a publicly available dataset released by Kuaishou. All data is collected from real user interaction records on the Kuaishou app, a popular short video app, between July 5 and September 5, 2020. It includes information such as video IDs, user IDs, video duration, video playback timestamps, and so on, thereby representing real user behavior when watching videos. We use the user interaction records from August 6, 2020, as the dataset for the experiment. It includes 117,977 video viewing records from 1,398 users requesting 10,230 video files, during 24 hours. In the experiment, we use the order of watched videos in the dataset as the original recommendation queue. However, due to the *Recommendation Reorder algorithm*, the order and timing of video views by users may not follow the original data, still ensuring video playback completion.

**Network conditions:** The experiment consists of two end-to-end network scenarios, i.e., from the cloud server to the base station (cable network) and from the base station to the end client devices (wireless network). We used the network traces provided by the MAWI Working Group [45] for the wide backbone network to simulate network throughput variations. Specifically, we use the network traces from August 12, 2020, from the main IX link of

WIDE to DIX-IE. The network fluctuations ranged from 60Mbps to 200Mbps in mean value, constructing different bandwidth scenarios (abundant bandwidth or limited bandwidth). We also used the FCC18 network traces to simulate network throughput fluctuations from the base station to the end client devices. The FCC18 dataset has been commonly used in previous works [53, 54]. To reflect the differences among clients, the network fluctuations were adjusted to differentiate bandwidth conditions including mean values of 2Mbps, 4Mbps, 6Mbps, and 12Mbps, each with a random variation of up to 5%.

**Video representations:** We use videos downloaded from the TikTok app and choose the corresponding video with a matching duration for each video in the dataset. Following the existing method [44], we divided each video into chunks of 1MB in size (the last chunk may be smaller than 1MB) and stored them on the cloud server. During the transmission process, if NCTM is triggered, these 1MB-sized chunks are XOR-encoded into coded files and transmitted to the clients over the network.

**Evaluation metrics:** NCTM was evaluated in terms of network bandwidth utilization, buffer variation, and rebuffer events. Network bandwidth utilization refers to the amount of bandwidth used between a cloud server and a base station. As a video is played, buffer variation refers to changes in the duration of the buffered content. Rebuffer events occur when video playback is interrupted due to low throughput. Since we consider the limited bandwidth struggles to meet the demands of all users, the coded transmission implemented by NCTM can transmit more data content within the limited bandwidth, thus reducing rebuffering events.