# OpenReview forum: "NCTM: A Novel Coded Transmission Mechanism for Short Video Deliveries"
_ACM.org/TheWebConf/2024/Conference — TheWebConf24_

### Official Review · Reviewer_mGmY · 2023-11-04

**Novelty:** 7
**Technical Quality:** 7

**Review:**

The paper presents, **NTCM**, a novel transmission of short videos as coded data instead of video content. **NTCM** XOR encodes the videos requested by multiple clients while ensuring successful decoding on each client by making effective use of user-side cached (watched) videos. **MTCN** is implemented using a novel algorithm (Minimum Clique Coverage algorithm) that is developed based on the concept of “cliques” and clique cover problem. This algorithm assists in efficiently encoding multiple requested videos by a group of clients, thus reducing the bandwidth demand. The paper also introduces a Recommendation Reorder algorithm, which reorders the recommended list of videos to create more opportunities for coded transmission. The system is evaluated comprehensively using real-life datasets, and a testbed of CDN, edge, and clients. **MTCN** reduces the bandwidth demand, and re-buffering events (time and number), compared to traditional CDN delivery and edge caching approaches.

**Pros:**
- The paper is well-written and organized and adds a significant amount of new knowledge.
- The system design and methodology are sound.
- The authors present an innovative system that creatively applies mathematical problems to a popular real-world application and solves critical issues, i.e., the high bandwidth demand of streaming short videos.
- The evaluation is comprehensive with suitable metrics, figures, and tables that enhance the clarity of the findings.
- The system performs very well in sufficient and limited bandwidth conditions and presents superior performance to edge caching and conventional CDN delivery.

**Cons:**
- Although the limitations of the SOTA were discussed very well in the intro, the research gap can be discussed and improved in the related works section. Currently, the motivation seems weak in the last paragraph of this section.

**Questions:**

- I would recommend discussing the research gap in the related works
- What is the difference between Figure 12 and Figure 13? Does each figure present the rebuffer request proportion in different settings of bandwidth?

**Ethics Review Description:**

No ethical issues

**Reviewer Confidence:**

4: The reviewer is certain that the evaluation is correct and very familiar with the relevant literature

**Scope:**

4: The work is relevant to the Web and to the track, and is of broad interest to the community

---

### Official Review · Reviewer_NGsN · 2023-11-21

**Novelty:** 5
**Technical Quality:** 7

**Review:**

The paper proposes a new transmission mechanism that harnesses the advantages of XOR codes and client-side caching for short video streaming. NCTM primarily consists of three components: the management of client cache information, the matching of client video queues for optimization opportunities, and the XOR encoder.

Pros:

1. The paper is well written and does a great job analyzing existing problems and data in short video streaming.

2. It introduces XOR codes for the use of short video data, and applies a novel mechanism to reduce the bandwidth usage.

3. The design of Minimum Clique Coverage algorithm is interesting, though it seems to not be able to guarantee a feasible solution in limited iteration or time.

Cons:

1. Most of the paper has been focus on analyzing and evaluating the effectiveness of algorithms for "Coded Chances Exploration", but the motivation to use of XOR codes for short video data has not been well addressed and tested. My concern is that the actual performance of XOR codes has largely to do with the characteristics of video contents and video codecs. The authors need to provide more information and reasons for this.

2. The frequent updating of clients and the video queue may lead to constant computing overhead for the cloud server. Additionally, the update latency from mobile clients to edge nodes and ultimately to the cloud server may cause untimely decisions for NCTM, resulting further performance degradation of the system.

3. As mentioned in Appendix C, NCTM is not suitable for situations with frequent handovers between base stations or frequent updates of client playbacks, such as watching progress and skip-overs. This poses a significant limitation for short video streaming, considering that many users consume videos on their phones while commuting in cars, buses, or subways.

4. The last sentence in paragraph 4 of Appendix B appears to be incomplete.

**Questions:**

Please refer to the cons.

**Reviewer Confidence:**

3: The reviewer is confident but not certain that the evaluation is correct

**Scope:**

4: The work is relevant to the Web and to the track, and is of broad interest to the community

---

### Official Review · Reviewer_oYcd · 2023-11-23

**Novelty:** 4
**Technical Quality:** 3

**Review:**

This paper proposes NCTM which is a caching technique designed for short form TikTok like videos. They key idea behind NCTM is to leverage similarity of videos between users to create and transmit a single file which is generated through an XOR operation on a set of videos. Users receive the XOR version and retrieve their requested video by running XOR operations using the videos individually cached by them. In order to make it work, the paper maintains history of each user in a user cache table and performs user group through a minimum coverage algorithm. Evaluations against LRU cache shows that NCTM is able to reduce peak bandwidth needs and reduces the number of rebuffer events and duration.

Pros:
- TikTok like short form videos are increasing in popularity
- CDNs continue to be interested in performant caching algorithms

Cons:
- Aspect of design are not fully explained
- Overheads associated with NCTM design are not evaluated thoroughly
- XOR coded caching has been proposed before.

**Questions:**

## Design

NCTM requires significant synchronization and collaboration between user requests, however, there is very little evidence presented in the paper that opportunities for such a collaborative technique exists. Given the availability of real traces, the paper should furnish a measurement study to motivate that opportunities for its design.

The paper makes a case that short videos are harder for CDN to effectively cache, because users have different preferences and thus the access pattern is random. I’m not sure if short videos are dissimilar to any other long tail workloads such as long form VOD or any other popularity driven content. The paper needs to make a clearer case to differentiate short videos from other web workloads.

For grouping to work well, there are two assumptions central to the design. First, it needs users to have complementary playback queues (perhaps this can be achieved with Recommendation Reorder). Second, it requires the user requests arrive very close to each other, it is unclear how this synchronization can be achieved in practice.

How does NCTM handle the following cases:
- A newly released short video which has not been watched before?
- First video requested by a user after opening their app (such that no local cached videos exist)
It seems that NCTM will not only need to cache original videos but also cache XOR versions.

## Evaluation
The evaluation seems to mostly focus on bandwidth reductions and reduced rebuffer times, however, this is of secondary importance. First the paper needs to show how hitrates under NCTM compare against baselines for a range of cache sizes.

Second, in evaluations shown, the baseline caching algorithm (CDN delivery mode, edge caching mode) is vanilla LRU. LRU is not the strongest baseline, there has been significant work on caching algorithms and I recommend the paper to compare against stronger baselines such as GDS/GDSF, LFU-DA, LHD, AdaptSize and AViC etc. A good list of video caching algorithm can be obtained from AViC: https://dl.acm.org/doi/abs/10.1145/3359989.3365423

The claim of sufficient time complexity in Sec 5.3 is not at all convincing. The paper needs to show the total time taken to handle an individual GET request and whether that is fast enough to handle highly concurrent workloads.

What is the memory overhead of user cache table and furthermore what is the compute complexity of keeping this table updated? This is an important aspect of NCTM design and must be devoted attention in evaluation.

**Reviewer Confidence:**

3: The reviewer is confident but not certain that the evaluation is correct

**Scope:**

3: The work is somewhat relevant to the Web and to the track, and is of narrow interest to a sub-community

---

### Official Review · Reviewer_xFeW · 2023-11-23

**Novelty:** 4
**Technical Quality:** 3

**Review:**

The paper aims at reducing the network overheads of streaming of short videos over the Internet. The key idea is to cache (some of) the videos that users already watched at the end devices, encode multiple videos in the same response, and decode the response given the cached videos. The goal of the paper is to maximize the number of encoded videos in a response while satisfying all user requests. The problem is transformed to a minimum clique coverage problem that the paper addresses. The paper also introduces a mechanism to modify the order of videos shown to users to leverage the previous algorithm. The paper evaluates the system using trace-driven simulations.


Strengths
- The caching, encoding and decoding approach is intuitive and novel to some extent.
- The paper is generally well-written.

Weaknesses
- Although the approach looks promising and intuitive, the whole solution doesn't seem to be practical:
  - For the video response to be sent, multiple users need to request a specific set of videos. This requires a synchronization "primitive" between *all* clients and the server, which is hard to achieve and is not scalable.
  - The paper doesn't discuss how broadcast is actually done. Assuming cellular networks, base stations need to set up multicast groups, and user equipment need to synchronize to receive contents of the resource blocks. Both issues are not evaluated or addressed in the paper. Similar challenges of multicast appear in other networks as well.
  - The solution doesn't address that a single chunk is often encoded to multiple bitrates.
  - It is not clear whether the proposed minimum clique coverage algorithm may result in fast churns or instabilities in the graph.
- Evaluation:
  - The evaluation is based on simulations. It is not clear whether the system will perform effectively in real deployments.
  - Many simulation details are missing such as implementation details of synchronization, broadcast etc. Also, the request rate, network topology, CDN caching policies etc.
  - The evaluation simulates ~10K videos and around 1400 users. These numbers are low given the high popularity of short videos noted by the paper. Also, this setup doesn't stress the proposed system in terms of performance gains and costs.
  - The paper doesn't explain how the network load savings are achieved and how much the broadcast/multicast contribute to these savings.


General comments:
- The term "relatively optimal solution" is not scientifically accurate.
- References [16] and [17] do not necessarily back up the claims made by the paper. For example, it seems that [17] focuses mostly on the impact of COVID-19's lockdowns on video and other traffic, and not necessarily related to short videos.
- In Figure 9, the paper mentions that the average network load decreases by 14%. CDFs cannot be used to report averages for general distributions.

**Questions:**

- Can you provide more details about the simulation? And why does it represent real deployments?
- What are the results of running the system in large-scale setups?
- How is broadcast/multicast implemented and realized?
- Can you address or elaborate on the need of synchronization discussed above?

**Reviewer Confidence:**

3: The reviewer is confident but not certain that the evaluation is correct

**Scope:**

3: The work is somewhat relevant to the Web and to the track, and is of narrow interest to a sub-community

---

### Official Review · Reviewer_ceV6 · 2023-11-30

**Novelty:** 4
**Technical Quality:** 5

**Review:**

The paper introduces NCTM, a system that aims to reduce the bandwidth strain caused by the rising popularity of short-form video. NCTM uses XOR-coded data transmission, leveraging cached data on user devices for encoding and decoding. NCTM is shown to reduce network load, peak traffic, and rebuffering events, outperforming traditional CDN and edge caching.

The paper is technically sound, with a detailed description of the NCTM architecture and principles, indicating a thorough approach in system design. The methodology, using trace-driven simulations for evaluation, is rigorous, though real-world testing would have added to the paper’s convincingness.

The paper has a well-organized structure and a clear exposition of technical concepts. However, the related works section is not as comprehensive as it could be, particularly with regard to short-form video delivery.

The paper demonstrates originality through the application of existing techniques, specifically coded caching, to the context of short-form video delivery. However, the field itself is quite niche, and while the approach is somewhat novel, it does not mark a substantial departure from existing work.

One drawback is the lack of clarity in Appendix C regarding the simplification of treating video chunks in adaptive bitrate streaming as distinct videos. This oversight raises questions about the applicability and scalability of NCTM.

To enhance the paper, a more extensive literature review and a deeper discussion on the implications of treating video chunks as distinct videos are advised.

In summary, the paper presents a well-conceived system for short-form video delivery. However, its contribution to the broader field is somewhat limited, and there are areas, particularly with regard to real-world applicability, where improvements can be made to strengthen the paper’s relevance and impact.

**Questions:**

* Is NCTM adaptable to other types of video content beyond short-form video?
* How does NCTM affect the quality of experience in terms of video playback and latency?
* How does NCTM scale in larger, more diverse network environments, and what challenges might arise in such scenarios?
* Can you elaborate on the limitations or implications of treating video chunks in adaptive bitrate streaming as distinct videos?
* Are there plans to conduct real-world testing to complement the trace-driven simulations and validate the practical applicability of NCTM?

**Reviewer Confidence:**

3: The reviewer is confident but not certain that the evaluation is correct

**Scope:**

3: The work is somewhat relevant to the Web and to the track, and is of narrow interest to a sub-community

---

### Decision · Program_Chairs · 2024-01-22

**Decision:**

Accept

**Comment:**

I summarise the pros and cons of the paper as follows.

 Overall Pros:
 Well-written and organized paper.
 Thorough evaluation with suitable metrics.
 Innovative system design that creatively applies mathematical problems.
 Superior performance compared to existing solutions.
 Clear presentation of findings.

 Overall Cons:
 Lack of real-world testing and evaluation.
 Lack of comprehensive literature review and discussion of research gap.
 Lack of clarity and thorough explanation in certain sections.
 Potential limitations and scalability issues of the proposed system.
 Lack of justification for the use of XOR codes for short video data.

 Most Major Issue:
 The most major issue in the reviews is the lack of real-world testing and evaluation. this may affect the overall validity and applicability of the proposed system.